# Occurrence of Multidrug-Resistant *Staphylococcus aureus* among Humans, Rodents, Chickens, and Household Soils in Karatu, Northern Tanzania

**DOI:** 10.3390/ijerph18168496

**Published:** 2021-08-11

**Authors:** Valery Silvery Sonola, Gerald Misinzo, Mecky Isaac Matee

**Affiliations:** 1Department of Wildlife Management, College of Forestry, Wildlife and Tourism, P.O. Box 3073, Morogoro 67125, Tanzania; 2Livestock Training Agency (LITA), Buhuri Campus, P.O. Box 1483, Tanga 21206, Tanzania; 3Department of Veterinary Microbiology, Parasitology and Biotechnology, College of Veterinary Medicine and Biomedical Sciences, Sokoine University of Agriculture, P.O. Box 3297, Morogoro 67125, Tanzania; gerald.misinzo@sacids.org; 4SACIDS Africa Centre of Excellence for Infectious Diseases, Sokoine University of Agriculture, P.O. Box 3297, Morogoro 67125, Tanzania; mateemecky@yahoo.com; 5Department of Microbiology and Immunology, Muhimbili University of Health and Allied Sciences, P.O. Box 65001, Dar es Salaam 11103, Tanzania

**Keywords:** *Staphylococcus aureus*, antibiotic resistance, humans, chickens, rodents, soil

## Abstract

We conducted this study to investigate the isolation frequency and phenotypic antibiotic resistance pattern of *Staphylococcus aureus* isolated from rodents, chickens, humans, and household soils. Specimens were plated onto mannitol salt agar (Oxoid, Basingstoke, UK) and incubated aerobically at 37 °C for 24 h. Presumptive colonies of *S. aureus* were subjected to Gram staining, as well as catalase, deoxyribonuclease (DNAse), and coagulase tests for identification. Antibiotic susceptibility testing was performed by using the Kirby–Bauer disc diffusion method on Mueller–Hinton agar (Oxoid, Basingstoke, UK). The antibiotics tested were tetracycline (30 μg), erythromycin (15 μg), gentamicin (10 μg), ciprofloxacin (5 μg), clindamycin (2 μg), and amoxicillin-clavulanate (20 μg/10 μg). The *S. aureus* strain American Type Culture Collection (ATCC) 25,923 was used as the standard organism. We found that 483 out of 956 (50.2%) samples were positive for *S. aureus*. The isolation frequencies varied significantly between samples sources, being 52.1%, 66.5%, 74.3%, and 24.5%, respectively, in chickens, humans, rodents, and soil samples (*p* < 0.001). *S. aureus* isolates had high resistance against clindamycin (51.0%), erythromycin (50.9%), and tetracycline (62.5%). The overall prevalence of multidrug-resistant (MDR) *S. aureus* isolates was 30.2%, with 8.7% resistant to at least four different classes of antibiotics.

## 1. Introduction

*Staphylococcus aureus* is both an opportunistic pathogen and a commensal microbe that colonizes a wide range of hosts, including humans, livestock, wild ungulates, and the environment [1,2,3]. *S. aureus* is also a leading cause of different infections in humans that range from minor skin infections to life-threatening diseases, such as pneumonia, osteomyelitis, endocarditis, and sepsis [2,4,5]. In farm animals, *S. aureus* causes mastitis in dairy animals [6,7] and septic arthritis in chickens [8], resulting in economic losses due to mortality and reduced production [9]. The pathogenicity of *S. aureus* is influenced by two important features: its ability to resist more than three classes of antibiotics [10] and the capacity to produce several toxins [2,11]. Multidrug-resistant bacteria have increased worldwide, resulting in the sharing of their genes with commensal microorganisms in humans, animals, and the environment and endangering public health [12]. Rodents have been extensively documented to carry and transmit different zoonotic pathogens, including *S. aureus*, to humans and livestock [3,13,14]. Commensal rodents colonized with pathogens have been widely reported to invade chicken [15,16] and human houses [17,18,19], exposing them to bacterial infections. Different studies in Tanzania have documented the interaction of rodents with humans in households, predisposing them to rodent-borne zoonotic diseases [20,21,22]. Rodent infestation in human settlements has been frequently reported in Karatu, where interactions of rodents with humans and livestock are very common, making it a plague focus area [20,21,23,24,25]. However, studies on the occurrence and pattern of multidrug-resistant (MDR) *S. aureus* among humans, rodents, and the environment in the area are missing. Therefore, this study aimed to determine the occurrence of MDR *S. aureus* isolates in humans, rodents, chickens, and soils in the households of Karatu in northern Tanzania.

## 2. Materials and Methods

### 2.1. Study Area

The study was conducted in the Karatu district in the northern zone of Tanzania between June 2020 and March 2021. Karatu is located between latitudes 3°10′ and 4°00′ S and longitude 34°47′ to 59.99′ E. The district has a population of 230,166 people comprised of 117,769 men and 112,397 women, with an average of five people per household. Karatu has an altitude range of 1000 to 1900 m above sea level with two wet seasons annually (short rains between October and December and long rains from March to June).

### 2.2. Sampling Strategy

The study population comprised of households keeping local chickens, while the sampling frame was the list of these households. Five wards, Karatu, Endabash, Endamarariek, Mbulumbulu, and Rhotia, were purposively selected based on the population density (at least 16,000 people), number of households with chickens, and household size of at least five people. Households were randomly selected from a list provided by a livestock field officer at the ward level by using a table of random numbers. At the household level, permission from the head of the household was granted first before trapping the rodents where areas for trapping in the surrounding environments relied on signs of rodents’ activities. For each household, one adult human (18 years and above) and one mature (seven months) scavenging chicken were involved in microbiological sampling to get one nasal swab and one cloaca swab, respectively. Furthermore, at least one rodent (in-house rat, peri-domestic rat, or both) could be captured, and one soil sample was collected per household. The selection of adult humans and mature chickens was based on the assumption that old individuals have been exposed to the interaction with rodents for a longer time than young ones, and hence are more likely to facilitate the sharing of infections.

### 2.3. Trapping of Rodents for Sample Collection

Live trapping of rodents was carried out using modified Sherman traps baited with peanut butter. An average of 100 traps (50 in houses and 50 in outside environments) were deployed per trap night for five consecutive nights in each ward. Each captured rodent was subjected to humane killing by using di-ethyl-ether and deep pharyngeal swabs, and the intestines were aseptically collected from the carcasses.

### 2.4. Collection of Samples from Humans, Chickens, and Soil

A total of 956 samples were collected from 286 households in the Karatu district wards. Of these, 286 were from chickens, 284 from humans, and 285 from soil (Table 1). Sterile cotton swabs were used to collect cloaca swabs from randomly picked scavenging chickens and human nasal swabs in households. Soil samples were randomly collected from five points in the household yards and mixed to compose one pooled soil sample [26]. Thereafter, cloaca and human nasal swabs were stored in sterile containers at −4°C and transported using Cary Blair transport medium and trypticase soy broth medium (Oxoid, Basingstoke, UK), respectively, to the Tanzania Veterinary Laboratory Agency (TVLA)–Arusha laboratory for processing within four hours after collection.

### 2.5. Culture, Isolation, and Identification of S. aureus Isolates

Specimens were plated onto mannitol salt agar (Oxoid, Basingstoke, UK) and incubated aerobically at 37 °C for 24 h. Presumptive colonies of *S. aureus* were subjected to Gram staining, as well as catalase, deoxyribonuclease (DNAse), and coagulase tests for identification.

### 2.6. Antibiotic Susceptibility Testing of S. aureus Isolates

An antibiotic susceptibility test was performed by using the Kirby–Bauer disc diffusion method on Mueller–Hinton agar (Oxoid, Basingstoke, UK) with commercially available discs, as described by [27]. The antibiotics tested were tetracycline (30 μg), erythromycin (15 μg), gentamicin (10 μg), ciprofloxacin (5 μg), clindamycin (2 μg), and amoxicillin-clavulanate (20 μg/10 μg). Pure colonies of the identified lactose fermenters were emulsified into 5 mL of sterile saline. The suspensions were adjusted to achieve a turbidity equivalent to 0.5 McFarland standard solutions, emulsified using sterile cotton swabs onto a Mueller–Hinton agar plate, and incubated at 37 °C for 16 to 18 h. After incubation, the inhibition zone of each antimicrobial agent was measured, and the results were interpreted according to the standards of [27]. *S. aureus* strain American Type Culture Collection (ATCC) 25,923 was used as the standard organism. An isolate was considered to be multidrug-resistant (MDR) if it was non-susceptible to three or more drugs from different classes of antibiotics [28].

### 2.7. Statistical Analyses

Isolation frequencies of *S*. *aureus* and the antibiotic resistance pattern of isolates were entered into Microsoft Excel version 2010 (Microsoft Corporation, Redmond, WA, USA) and their percentages were calculated by descriptive statistics. The association between categorical variables was analysed by using a chi-squared (Fisher’s exact and Pearson’s) test. Statistical significance was accepted at *p <* 0.05.

## 3. Results

### 3.1. Isolation of Staphylococcus aureus from the Samples

Overall, 483 samples out of 956 (50.5%) had *S. aureus*. Significant variation in isolation frequencies was observed between the types of samples, being higher in rodents (74.3%) compared to soil (24.5%) samples (*p* < 0.001) (Table 1).

**Table 1 ijerph-18-08496-t001:** Isolation frequencies of *Staphylococcus aureus* from different sample sources.

Types of Sample Sources	Number of Samples *n* (%)	Positive Samples *n* (%)	Chi-Squared	*p*-Value
Chickens	286 (29.9)	149 (52.1)	X^2^ = 83.849, df = 3	<0.001
Humans	284 (29.7)	189 (66.5)		
Rodents	101 (10.6)	75 (74.3)		
Soil	285 (29.8)	70 (24.5)		
Total	956 (100.0)	483 (50.5)		

### 3.2. Antibiotic Susceptibility Testing (AST) Results of the S. aureus Isolates

The overall resistance rates were 51.0% to clindamycin, 50.9% to erythromycin, 6.9% to ciprofloxacin, 62.5% to tetracycline, 2.2% to gentamycin, and 10.7% to amoxicillin-clavulanate. The specific resistance rates are shown in Table 2 and Figure 1.

### 3.3. Prevalence of Multidrug-Resistant Isolates of S. aureus in Different Types of Samples

About 146 out of 483 isolates (30.2%) were resistant to at least three different classes of antibiotics. The population of MDR *S. aureus* was composed of 70 (14.5%), 51 (10.6%), 15 (3.1%), and 10 (2.1%) isolates from chicken, human, rodent, and soil samples, respectively (Table 3). The MDR rates varied significantly for isolates from chickens, humans, rodents, and soil (*p* < 0.001). In all types of samples, none of the MDR *S. aureus* isolates were resistant to all six classes of antibiotics.

### 3.4. Prevalence of MDR S. aureus in Samples from Different Wards in the Study Area

Most of the MDR *S. aureus* isolates (43.2%) were found in samples from Endamarariek, followed by Karatu (21.9%) and Endabash (15.8%), while a few MDR isolates were observed in samples from Mbulumbulu (11.0%) and Rhotia (8.2%) (Figure 2). The occurrence of MDR isolates varied significantly in samples from the Endabash, Karatu, Endamarariek (*p* < 0.001), Mbulumbulu (*p* < 0.006), and Rhotia (*p* < 0.005) wards (Table 4).

### 3.5. Phenotypic Patterns of MDR S. aureus Isolates

As shown in Table 5, MDR *S. aureus* isolates displayed variable resistance patterns, where CD-E-TE was the most common that appeared in chicken (34.2%), human (14.4%), rodent (3.4%), and soil (1.4%) isolates. CD-E-TE-AMC was also common in chicken (6.2%), human (6.2%), and rodent (2.7%) isolates, but not in the soil isolates. Patterns showing resistance to five different classes of antibiotics were CD-E-TE-CN-AMC found in soil and rodent (0.7%) samples and CD-E-CIP-TE-AMC in soil (1.4%) and human (4.8%) samples. However, none of the MDR *S. aureus* isolates were resistant to all antibiotic classes.

## 4. Discussion

This is the first study to investigate the carriage of *S. aureus* in chickens, humans, rodents, and soils in a household environment in Tanzania. Overall, the isolation frequency of *S. aureus* was 50.5%. We observed significant variations in isolation frequencies among sample sources, where rodents had more *S. aureus* (74.3%), and soil had the lowest (24.5%). The presence of drug-resistant bacteria in soil serves as a potential reservoir of antibiotic resistomes, which encompasses all types of antibiotic resistance genes (ARGs) that can spread to humans and animals and to a wider environment [29,30]. Rodents carrying different zoonotic pathogens have been frequently reported to invade human residences in Karatu [20,21,22]. Overall, the isolates exhibited high resistance to clindamycin (51.0%), tetracycline (62.5%), and erythromycin (50.9%). These antibiotics are commonly used in humans and poultry production in the study area, and their frequent use and misuse can significantly contribute to increased resistance [6,18,31,32,33]. In this community, there is frequent use and misuse of the drugs in food animals, including poultry, mainly tetracycline and erythromycin [34,35]. Farmers in rural areas of Tanzania have been treating their chickens with antibiotics without diagnosis or prescriptions from veterinarians [35].

Our study observed that 146 out of 483 (30.2 %) isolates were MDR, including 14.5% chicken, 10.6% human, 3.1% rodent, and 2.1% soil isolates. The higher prevalence of MDR *S. aureus* in humans and poultry can be associated with the extensive use of drugs in human medicine and poultry in the community [36]. Lower multidrug resistance rates in rodents could be because these animals are not direct consumers of antibiotics, as is the case for humans and chickens. Their exposure to drugs is indirect, depending on contact with human and chicken wastes when dropped in the household environment, as explained in other studies [37,38]. Our findings are in keeping with those of Vitale et al. [32], showing that *S. aureus* derived from humans were more resistant to antibiotics compared with those of animal origin.

In our study, most MDR isolates were found in the Endamarariek ward (12.6%), which is basically a rural area compared to Karatu (6.6%), an urban and district headquarter. These variations could be due to differences in the levels of awareness and use of antibiotics between the wards. Endamarariek is a rural area with a scarcity of veterinarians, where farmers mostly treat their chickens based on experiences using home-stored antibiotics and those purchased from village shops with a low level of control. Such variations can also explain why we found more MDR *S. aureus* isolates (2.1%) in rodent samples from Endamarariek compared to Karatu samples (0%). Among MDR patterns, CD-E-TE, standing for clindamycin, erythromycin, and tetracycline, was displayed in most of the isolates (34.2 %), and the pattern is identical for humans, chickens, and rodents. Different studies on the resistance profiles of *S. aureus* have reported similar patterns as well [7,18,32,33,39,40]. Erythromycin and tetracycline are the most commonly used antibiotics in this area, since they are cheap and can be purchased over the counter without a prescription [41].

### Limitation of the Study

Despite our findings being useful in the control of antimicrobial resistance in Tanzania, a genetic characterization of antibiotic resistance and virulence factors of *S. aureus* could provide additional information to compliment the phenotypic approach.

## 5. Conclusions

These results suggest a potential role of the interaction of humans, chickens, and rodents in cross-transmission of MDR *S. aureus* among them, with the possibility of causing human and animal infections that are difficult to treat. Unfortunately, treatment alternatives are very limited due to the few types of antibiotics in the studied area and the economic reality. Therefore, necessary interventions, such as continuous educative campaigns on effective cleanliness in households, safe disposal of animal wastes, and rodent control strategies, are urgently needed.

## Figures and Tables

**Figure 1 ijerph-18-08496-f001:**
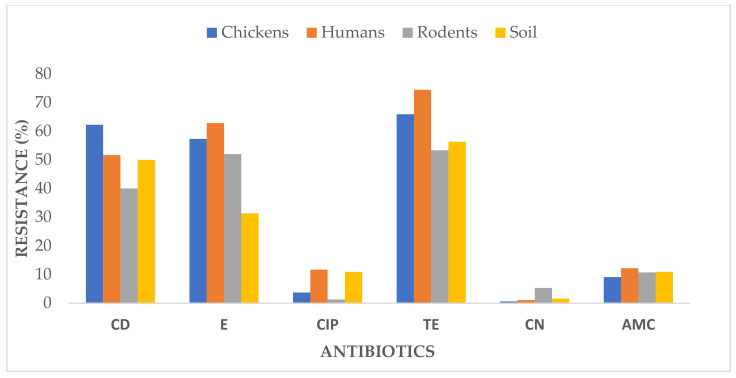
Resistance of *S. aureus* isolates against the antibiotics; CD = clindamycin, E = erythromycin, CIP = ciprofloxacin, TE = tetracycline, CN = gentamycin, and AMC = amoxicillin-clavulanate.

**Figure 2 ijerph-18-08496-f002:**
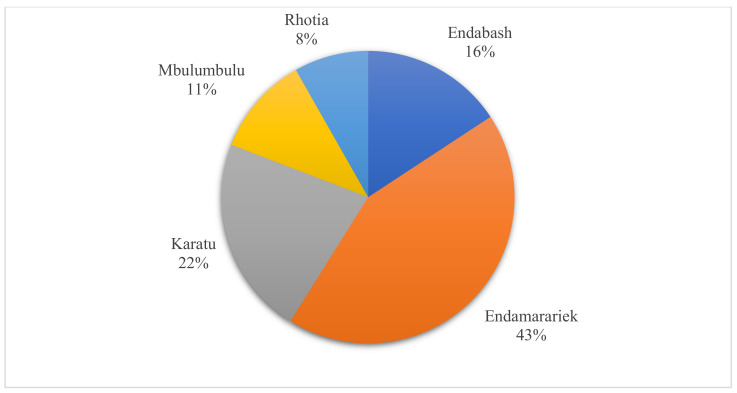
Distribution of MDR *S. aureus* isolates in different wards of Karatu.

**Table 2 ijerph-18-08496-t002:** Antibiotic resistance pattern of *Staphylococcus aureus* isolates from chicken, human, rodent, and soil samples.

Sample Type	Antibiotics, *n* (%)						Overall R	Chi-Squared Test
	Clindamycin	Erythromycin	Ciprofloxacin	Tetracycline	Gentamycin	Amoxicillin-Clavulanate		
Overall R	51.0 %	50.9 %	6.9 %	62.5 %	2.2 %	10.7 %		
Chickens								
R	102 (62.2)	94 (57.3)	6 (3.7)	108 (65.9)	1 (0.6)	15 (9.1)	33.1 %	247.61, df = 5, *p* < 0.001
I	16 (9.8)	21 (12.8)	17 (10.4)	13 (7.9)	0 (0.0)	2 (1.2)		
S	46 (28.0)	49 (29.9)	141 (86.0)	43 (26.2)	163 (99.4)	147 (89.6)		
Subtotal	164 (100.0)	164 (100.0)	164 (100.0)	164 (100.0)	164 (100.0)	164 (100.0)		
Humans								
R	93 (51.7)	113 (62.8)	21 (11.7)	134 (74.4)	2 (1.1)	22 (12.2)	35.7 %	243.1, df = 5, *p* < 0.001
I	12 (6.7)	16 (8.9)	18 (10.0)	13 (7.2)	9 (5.0)	6 (3.3)		
S	75 (41.7)	51 (28.3)	141 (78.3)	33 (18.3)	169 (93.9)	152 (84.4)		
Subtotal	180 (100.0)	180 (100.0)	180 (100.0)	180 (100.0)	180 (100.0)	180 (100.0)		
Rodents								
R	30 (40.0)	39 (52.0)	1 (1.3)	40 (53.3)	4 (5.3)	8 (10.7)	27.1 %	79.74, df = 5, *p* < 0.001
I	10 (13.3)	11 (14.7)	8 (10.7)	9 (12.0)	1 (1.3)	5 (6.7)		
S	35 (46.7)	25 (33.3)	66 (88.0)	26 (34.7)	70 (93.3)	62 (82.7)		
Subtotal	75 (100.0)	75 (100.0)	75 (100.0)	75 (100.0)	75 (100.0)	75 (100.0)		
Soil								
R	32 (50.0)	20 (31.3)	7 (10.9)	36 (56.3)	1 (1.6)	7 (10.9)	26.8 %	61.21, df = 5, *p* < 0.001
I	6 (9.4)	7 (10.9)	4 (6.3)	6 (9.4)	0 (0.0)	1 (1.6)		
S	26 (40.6)	37 (57.8)	53 (82.8)	22 (34.4)	63 (98.4)	56 (87.5)		
Subtotal	64 (100.0)	64 (100.0)	64 (100.0)	64 (100.0)	64 (100.0)	64 (100.0)		

R = resistant, I = intermediate, and S = susceptible.

**Table 3 ijerph-18-08496-t003:** MDR rates of *S. aureus* isolates from different types of samples.

Type of Sample Source	Number of Antibiotic Classes to Which the Isolates Were Resistant, *n* (%)									Chi-Squared	*p*-Value
	0	1	2	3	4	5	6	Total Isolates	MDR Isolates (3–6 Classes)		
Overall	81 (16.8)	74 (15.3)	182 (37.7)	104 (21.5)	32 (6.6)	10 (2.1)	0 (0.0)	483 (100.0)	146 (30.2)		
Chickens	34 (42.0)	15 (20.3)	45 (24.7)	61 (58.7)	9 (58.7)	0 (0.0)	0 (0.0)	164 (34.0)	70 (14.5)	143.66 df = 3	*p* < 0.001
Humans	12 (14.8)	31 (41.9)	86 (47.3)	30 (28.8)	14 (43.8)	7 (70.0)	0 (0.0)	180 (37.3)	51 (10.6)	195.12 df = 3	*p* < 0.001
Rodents	19 (23.5)	16 (21.6)	25 (13.7)	8 (7.7)	7 (21.9)	0 (0.0)	0 (0.0)	75 (15.5)	15 (3.1)	51.47 df = 6	*p* < 0.001
Soil	16 (19.8)	12 (16.2)	26 (14.3)	5 (4.8)	2 (6.3)	3 (30.0)	0 (0.0)	64 (13.3)	10 (2.1)	57.84 df = 6	*p* < 0.001

**Table 4 ijerph-18-08496-t004:** Prevalence of MDR *S. aureus* isolates in different samples by wards.

Wards	MDR Isolates from Different Sample Sources *n* (%)					Chi-Squared	*p*-Value
	Chickens	Humans	Rodents	Soil	Total		
Overall MDR	70 (14.5)	51 (10.6)	15 (3.1)	10 (2.1)	146 (30.2)		
Endabash	16 (3.3)	3 (0.6)	3 (0.6)	1 (0.2)	23 (4.8)	24.826 df = 3	<0.001
Endamarariek	18 (3.7)	30 (6.2)	10 (2.1)	3 (0.6)	61 (12.6)	29.508 df = 3	<0.001
Karatu	20 (4.1)	8 (1.7)	0 (0.0)	4 (0.8)	32 (6.6)	28 df = 3	<0.001
Mbulumbulu	8 (1.7)	7 (1.4)	1 (0.2)	1 (0.2)	17 (3.5)	12.5 df = 3	0.0059
Rhotia	8 (1.7)	3 (0.6)	1 (0.2)	1 (0.2)	13 (2.7)	12.667 df = 3	0.0054
Chi-squared	9.1429	55.962	22	8.25			
*p*-Value	0.0576	<0.001	0.0002	0.0828			

**Table 5 ijerph-18-08496-t005:** Phenotypic resistance patterns of MDR *S. aureus* isolates from chickens, humans, rodents, and soil samples.

Source of Samples (*N* = 146)	Number of Isolates (*n*)	Occurrence (%)	Antibiotic Resistance Patterns	Number of Antibiotic Classes
Chickens	50	34.2	CD, E, TE	3
(*n* = 70)	3	2.1	CD, CIP, TE	
	1	0.7	E, CIP, TE	
	5	3.4	E, TE, AMC	
	9	6.2	CD, E, TE, AMC	4
	1	0.7	CD, E, CIP, TE	
	1	0.7	CD, E, TE, CN	
Humans	21	14.4	CD, E, TE	3
(*n* = 51)	3	2.1	CD, CIP, TE	
	2	1.4	CD, E, CIP	
	1	0.7	CD, E, AMC	
	1	0.7	CD, TE, AMS	
	1	0.7	E, TE, AMC	
	9	6.2	CD, E, TE, AMC	4
	1	0.7	E, CIP, TE, CN	
	3	2.1	CD, E, CIP, TE	
	2	1.4	CD, CIP, TE, AMC	
	7	4.8	CD, E, CIP, TE, AMC	5
Rodents	5	3.4	CD, E, TE	3
(*n* = 15)	1	0.7	CD, E, AMC	
	1	0.7	CIP, CN, AMC	
	1	0.7	CD, TE, AMC	
	4	2.7	CD, E, TE, AMC	4
	2	1.4	CD, E, TE, CN	
	1	0.7	CD, E, TE, CN, AMC	5
Soil	2	1.4	CD, E, TE	3
(*n* = 10)	1	0.7	E, TE, AMC	
	1	0.7	CD, TE, AMC	
	1	0.7	CD, CIP, TE	
	2	1.4	CD, E, CIP, TE	4
	2	1.4	CD, E, CIP, TE, AMC	5
	1	0.7	CD, E, TE, CN, AMC	
Total	146	100.0		

AMC = amoxicillin-clavulanate, TE = tetracycline, E = erythromycin, CD = clindamycin, CIP = ciprofloxacin, and CN = gentamycin.

## Data Availability

The data presented in this study are available on request from the corresponding author. The data are not publicly available due to privacy restrictions.

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
