# Peer review of "Occurrence of Multidrug-Resistant Staphylococcus aureus among Humans, Rodents, Chickens, and Household Soils in Karatu, Northern Tanzania"

_ijerph, 2021, doi:10.3390/ijerph18168496_

Round 1

Reviewer 1 Report

The manuscript is a good initiative, data collection as per methods seems ethical and the data is well presented. It suffers due to lazy analysis and predictable conclusions which makes the reader lose interest by the time one reaches the discussion section. The abstract doesn't capture the impact of the study. The discussion seems purely speculative with the majority of the conclusions not followed upon. The points for authors to consider are : 

  1. L22 - unclear - Resistant to at least four classes of?
  2. L32 - unclear- reduction production of? 
  3. L34-36 is confusing. Multidrug-resistant bacteria have increased sharing of "drug-resistant genes" and among "commensals, animals, humans and the environment" and how? 
  4.  l52 comprising of
  5. Section 2.2 line 58 - 71, to be re-written 
  6. Section 2.6 Line 92 - E. coli or S. aureus? 
  7. Section 3.1 Elaborate on the results? When you report the percentage of total isolates, are you losing information when you group various types of samples together? Would it make more sense to have the number of positive be represented as the % positive of the number of samples collected of that particular class to draw meaningful conclusions and state how the abundance of staph varies within different population groups? 
  8. Section 3.2 and discussion lines L172-173. The authors pin the occurrence of higher resistance isolates to the unregulated antibiotic use in animal husbandry and non-prescriptive use of these antibiotics in the human population of the region which is a plausible explanation. The results clearly indicate 3 drugs to be most important - Clindamycin, erythromycin, gentamycin but don't specify why these 3 stand out and not others. The two could be tied, but a detailed explanation in the discussion section would add value to the results. 
  9. Section 3.3, could authors offers an explanation for why rodents who have been living in these environments present low antibiotic resistance rates specifically as if one were to assess the transfer between all these species, an ideal relationship should be 1:1:1:1 for human: chicken: rodent: soil even if one sample is collected as these interactions in an environment with controlled intermixing would have given sufficient time for exposure to have occurred. Different rates could be expected if this habitat and these interactions are relatively newer.

Overall this study suffers from a lack of a control group where one of the species from the sample set has been removed to assess the contribution of each of these interactions to have more meaningful data honing on the exact mediators of inter-species pathogen transfer. 

Reviewer 2 Report

Authors reported the isolation frequency and phenotypic antibiotic resistance pattern of Staphylococcus aureus isolated from rodents, chickens, humans and household soils. They refers that the 50.2 % of samples were positive for S. aureus and that isolation frequencies was different in chickens, humans, rodents and soil samples (15.6 %, 19.8 %, 7.8 % and 7.3 % respectively). S. aureus isolates had high resistance against clindamycin (51.0%), erythromycin (50.9%) and tetracycline (62.5%).

The results obtained are interesting, but the manuscript must be improved.

As the authors themselves write, a limitation of the work is the lack of genetic characterization of antibiotic resistance and virulence factors of S. aureus. Check that in the entire text S. aureus is written in italic. The English version must be revised

Abstract

Authors must be insert some information about material and method and must be write S. aureus in italic

Keywords

I suggest to substitute “animals” with “chickens” and to delete “antibiotics” leaving “antibiotics resistance"

Material and method

2.1. Study area

Line 54-56: I suggest to delete this sentence. It doesn’t give useful information for the purpose of the study unless in the discussion environmental factors are linked with the results obtained

2.2. Sampling strategy

Line 58-59: The sentence is not clear

How many samples were collected? I suggest to insert a table with all specifications

2.3. Trapping of rodents for sample collection

Line 75-76: this sentence is superfluous for the purpose of the study. Even this sentence can have a meaning if contextualized in the discussion

2.6. Antibiotic susceptibility testing of E. coli isolates

There is probably a typo: S. aureus and not E. coli

Line 102: The authors take into consideration the clinical breakpoints compiled by CLSI 2020. It is more appropriate to take in consideration the clinical breakpoints compiled by EUCAST (http://www.eucast.org/clinical_breakpoints/) to define resistance/susceptibility (2021).

I would also add methicillin and vancomycin to the antibiotics tested. Why weren't they used?

Discussion

The discussion needs to be improved. I suggest authors consider the papers listed below:

  • Vitale M, Galluzzo P, Buffa PG, Carlino E, Spezia O, Alduina R. Comparison of Antibiotic Resistance Profile and Biofilm Production of Staphylococcus aureus Isolates Derived from Human Specimens and Animal-Derived Samples. Antibiotics (Basel). 2019 Jul 19;8(3):97. doi: 10.3390/antibiotics8030097. PMID: 31330991; PMCID: PMC6783831.
  • Amoako DG, Somboro AM, Abia ALK, Molechan C, Perrett K, Bester LA, Essack SY. Antibiotic Resistance in Staphylococcus aureusfrom Poultry and Poultry Products in uMgungundlovu District, South Africa, Using the "Farm to Fork" Approach. Microb Drug Resist. 2020 Apr;26(4):402-411. doi: 10.1089/mdr.2019.0201. Epub 2019 Oct 24. PMID: 31647362.
  • Oladipo AO, Oladipo OG, Bezuidenhout CC. Multi-drug resistance traits of methicillin-resistant Staphylococcus aureus and other Staphylococcal species from clinical and environmental sources. J Water Health. 2019 Dec;17(6):930-943. doi: 10.2166/wh.2019.177. PMID: 31850900.

Also, I would do a more careful analysis of results obtained to give more emphasis.

It should be emphasized that the most present combinations of antibiotic resistance are identical between humans, chickens and rodents.

Round 2

Reviewer 2 Report

Authors have improved their paper, but there are still revisions to be made that have not been answered.

Abstract

Authors must be insert some information about material and method and must be write S. aureus in italic

Keywords

I suggest to substitute “animals” with “chickens” and to delete “antibiotics” leaving “antibiotics resistance”

Material and method

How many samples were collected? How many families? In the material and method continues to be unclear  

2.3. Trapping of rodents for sample collection

"Morphometric measurements were recorded for each captured rodent"...It is information that has not followed neither in  results nor in discussion. I would eliminate this sentence

Discussion

Line 164: "We observed significant variations in isolation frequencies among sample sources, where rodents had more S. aureus (74.3%) and soil had the lowest (24.5%)."

I think it is important to comment on this result as well. In this regard, I suggest the following references:

  1. D'Costa VM, Griffiths E, Wright GD. Expanding the soil antibiotic resistome: exploring environmental diversity. Curr Opin Microbiol. 2007 Oct;10(5):481-9. doi: 10.1016/j.mib.2007.08.009. Epub 2007 Oct 22. PMID: 17951101.
  2. Woolhouse M, Ward M, van Bunnik B, Farrar J. Antimicrobial resistance in humans, livestock and the wider environment. Philos Trans R Soc Lond B Biol Sci. 2015 Jun 5;370(1670):20140083. doi: 10.1098/rstb.2014.0083. PMID: 25918441; PMCID: PMC4424433.
